# Experiences of Interdisciplinary Working from the Perspective of the Society of Master Saddlers Qualified Saddle Fitters

**DOI:** 10.3390/ani14040559

**Published:** 2024-02-07

**Authors:** Kathryn Nankervis, Russell MacKechnie-Guire, Christy Maddock, Alison Pyatt

**Affiliations:** 1Equine Department, Hartpury University, Gloucestershire GL19 3BE, UK; russell.mackechnie-guire@hartpury.ac.uk (R.M.-G.); christy.maddock2@hartpury.ac.uk (C.M.); 2International Office, Veterinary Medicines Directorate, Addlestone KT15 3LS, UK; a.pyatt@vmd.gov.uk

**Keywords:** horse, saddle, fitter, multi-disciplinary, equestrian, professional

## Abstract

**Simple Summary:**

An equestrian multi-disciplinary team (MDT) is a group of equestrian professionals providing advice for horse owners on matters relating to the care of their horse. The aim of this study was to explore the experiences of the Society of Master Saddlers qualified saddle fitters (SMSQSFs) working with other equestrian professionals within an MDT. Semi-structured, one-to-one online interviews with fourteen SMSQSFs (participants) were completed, providing information about the frequency and type of interactions with other professionals; horse owner expectations of an MDT approach; and any benefits, challenges, and barriers to an MDT approach within an equestrian setting. Interviews were video and audio recorded, transcribed, and imported into a qualitative data analysis software tool for thematic analysis. Six themes were identified: (1) effective communication; (2) multidisciplinary expectations; (3) horse welfare; (4) professionalism; (5) relationships; (6) working together. Most participants felt they could have a positive effect on horse welfare and horse–rider performance but were under-valued as a profession. The role of the horse owner within the MDT was unclear. Similarities exist between equestrian MDT and human healthcare teams. Equestrian professionals working effectively within an MDT can support horse owners in achieving optimal saddle fit.

**Abstract:**

Horse owners seek the advice and support of a number of equestrian professionals in carrying out their duty of care for their animal. In some instances, these professionals form a multi-disciplinary team (MDT). The aim of this study was to explore the experiences of the Society of Master Saddlers’ qualified saddle fitters (SMSQSFs) working with other professionals and to understand the nature of inter-disciplinary working from an SMSQSF perspective. Semi-structured, one-to-one online interviews with fourteen SMSQSFs were completed. Areas explored included the nature of the participant’s client base; the frequency and nature of their interactions with other professionals; their perceptions of horse owner expectations of an MDT approach; and any benefits, challenges, and barriers to an MDT approach within an equestrian setting. Interviews were video and audio recorded (MS Teams), transcribed verbatim (Otter ai), and imported into qualitative data analysis software (NVivo, version 12). Data were analysed using thematic analysis. Six themes were identified: (1) effective communication; (2) multidisciplinary expectations; (3) horse welfare; (4) professionalism; (5) relationships; (6) working together. Communication was recognised as a crucial component of an effective MDT. Most participants valued and desired an MDT approach. They felt they had a key role to play within the equestrian MDT, not only in the prevention of deterioration in horse welfare but also in improving the functionality and performance of the horse–rider partnership. Effective MDT working was also seen as having benefits to SMSQSFs and other professional stakeholders alike, although time and financial constraints were identified as barriers to MTD working. The role of the horse owner within the MDT was unclear and potentially complex, and this and other factors such as the professional identity of the SMSQSF, personal relationships, and input from others outside of the MDT team were identified as challenges to effective MDT working. This present study found that SMSQSFs experience similar benefits and challenges to an MDT approach as seen in human healthcare settings. The role of the horse owner, communication, and professional recognition are indicated as pivotal to MDT effectiveness in achieving optimal saddle fit.

## 1. Introduction

In the United Kingdom, there are approximately 331,000 horse-owning households with an estimated horse population standing at 726,000 [1]. Whether horses are used in sport, for leisure, or for breeding purposes, their welfare is the responsibility of those who own, breed, train, ride, and care for them. Horse owners seek the advice and support of a diverse range of equestrian professionals in carrying out their duty of care, and these professionals form a multi-disciplinary team (MDT). Within human healthcare settings the MDT is defined as a team of healthcare professionals, for example, general practitioners, social workers, and nurses, representing different organisations and professions working together to improve care and health outcomes [2,3]. In the absence of reliable evidence regarding the MDT approach in equine healthcare and, more widely, animal healthcare, extrapolation from the extensive and generally analogous human healthcare sector is necessary.

Within equine settings, the MDT structure is less well established but typically consists of a veterinary surgeon, farrier, and others such as a musculoskeletal therapist, coach, equine dental technician, nutritionist, and saddle fitter. This is not an exhaustive list but gives an indication of the complex landscape faced by the horse owner when working with equine health professionals. Whilst the benefits of a team approach to horse care have been recognised with respect to the relationship between the veterinary profession and others, such as farriers [4] and veterinary physiotherapists [5], there remains a paucity of evidence exploring the nature of the MDT within equestrian settings, the challenges faced by the horse owner, and the potential benefits for horse welfare and equestrian practices of effective MDT working.

One of the challenges facing horse owners is deciding which equestrian professionals form their MDT. To an extent, some of this decision making is facilitated by UK legislation, since the practice of certain members of the MDT is subject to legislation via the Veterinary Surgeons Act 1966 [6] and the Farriers (Registration) Act 2017 [7]. The Veterinary Surgery (Exemptions) Order (2015) [8] allows for people who are not registered in the Register of Veterinary Surgeons to carry out specified acts such as physiotherapy, with “Physiotherapy” interpreted as including all kinds of manipulative therapy. Whilst osteopaths, chiropractors, and physiotherapists fall under statutory regulation as regards their human practice, they are subject to voluntary regulation only regarding animal practice. Similarly, saddle fitting in the UK does not come under any statutory regulation, although the recently launched “Equine Fitters Council” (EFC) now provides voluntary regulation. The EFC aims to “set standards, promote education and encourage professional practice within the industry” [9].

The requirement for professional saddlery fitting is becoming more widely recognised as a result of a growing body of evidence indicating the importance of correct saddlery (saddle and bridle fit) for the welfare of the horse. Poor practice is seen as a risk to horse welfare [10,11,12]. Whilst there are various training pathways, the Society of Master Saddlers offer an industry-recognised and independently assessed qualification enabling individuals to train to become a Society of Master Saddlers’ qualified saddle fitter (SMSQSF) (see Table 1). SMSQSFs are subject to a Code of Professional Conduct which includes Professional Standards describing how they should interact with horses, their clients, and other professionals within an MDT.

Within a human healthcare setting, the benefits of MDT working are well recognised as improved patient care, better clinical outcomes [13], and shared knowledge and good practice [14]. Within equine healthcare, input from non-veterinary surgeons has been associated with the prevention of certain conditions, e.g., the farrier in the prevention of laminitis [15] and the veterinary physiotherapist in the prevention of pelvic and hindlimb fracture in racehorses [16]. It has been recommended that all those within the MDT team are aware of the association between saddle slip and musculoskeletal health, since saddle slip can be an indicator of lameness [17,18]. Whilst such studies intimate the value of an inter-disciplinary approach to equine healthcare, the subject of the equine MDT and the role of saddle fitters within the MDT in particular has not been explicitly addressed within the equine veterinary literature.

The aim of this present study was to capture and explore the experiences of SMSQSFs in inter-disciplinary working and to understand the current nature of interdisciplinary working from the perspective of the SMSQSF.

## 2. Materials and Methods

The study was approved by Hartpury University’s Research Ethics Committee (ETHICS 2020-07). All interviews were confidential and fully anonymised. Study participants gave full informed written consent prior to interview. All data were held in accordance with General Data Protection Regulation.

### 2.1. Study Design

Qualitative methodologies are increasingly recognised, valued, and applied within veterinary science research to gain in-depth insight and understanding of complex concepts or problems from the stakeholder perspective [19,20,21]; hence, a qualitative approach was adopted. Semi-structured, one-to-one interviews with SMSQSFs were completed to explore their opinion and experiences of interdisciplinary working across a range of equestrian and veterinary professionals including veterinarians, equestrian coaches, musculoskeletal practitioners (MSK), and farriers and saddle fitters (SMSQSF-qualified or otherwise). The topics for discussion were developed in accordance with the study aims (Table 2). The main questions and suggested probing questions/prompts for each main question formed the interview guide (Appendix A: Interview guide).

Once the interview questions were finalised, a call for participation was published, using the Society of Master Saddlers’ social media and circulation lists. Forty-four SMSQSFs volunteered to be interviewed, from which fourteen participants were selected by the research team for interview. Purposive sampling was used to ensure representation of a broad demographic in terms of the geographical base within the UK, number of years of saddle fitting experience, and type of business operation (limited company, sole trader partnership).

### 2.2. Data Collection

Interviews (*n* = 14) were undertaken over a 3-month period between December 2020 and February 2021, led by an experienced qualitative researcher (AZP). At the start of each interview, the interviewer took participants through a standard pre-interview checklist, clarifying the aims of the project, the nature of the technique to be used, gave reassurance regarding confidentiality and data protection, and reminded participants of their right to withdraw at any point.

All interviews adhered to the agreed interview guide (Appendix A: Interview Guide). Participants were allowed to talk freely about the topics raised for discussion and were also permitted the opportunity to clarify or make additional comments at the end of the interview. In accordance with the pre-determined questions, participants were asked to share their experience and opinions on a range of topics around multidisciplinary working practices. Interview duration was between 35 and 90 minutes. Interviews were held under conditions of strict confidence. Interview data were fully anonymised prior to analysis.

Interviews were video and audio-recorded using Microsoft Teams (Microsoft, Redmond, WA, USA). Interviews were transcribed (verbatim) using Otter ai software (Otter. ai, Inc., Mountain View, CA, USA) as a supportive technique. All transcripts were checked for accuracy, cleaned (CM), and imported into the qualitative data analysis (QDA) software NVivo, Version 12, QSR International Pty Ltd., Burlington, MA, USA) for analysis. The qualitative reporting process was undertaken in accordance with the consolidated criteria for reporting qualitative studies (COREQ) checklist [22] to support the methodological process, analysis, and interpretation of the data [23].

### 2.3. Thematic Analysis

The interview transcripts were analysed using the established qualitative research technique, thematic analysis, whereby researchers scrutinise and examine the dataset for common themes and patterns [24,25]. Iterative coding facilitated the organisation of interview data into meaningful topic groups, i.e., themes, to reflect the interviewee’s comments [26,27]. Concurrent data collection and analysis using the QDA software was completed by the research team, and the six-stage process as defined by Braun and Clark [26,27] followed: (1) transcripts were read and re-read repeatedly to enable familiarisation with the content prior to commencement of coding; (2) coding was conducted to generate concise labels for important features; (3) coherent and meaningful patterns in the codes were identified to enable the formation of themes; (4) themes were reviewed to reflect the data; (5) themes were defined and named with appropriate detailed analysis; (6) themes were contextualised against the existing literature and supporting evidence.

## 3. Results

The participants were SMSQSFs a minimum; five were also SMSMSFs and eight were SMSMSs. Eleven were self-employed and the remaining three were employed as fitters. Six consistent themes of equal importance were identified within the dataset; these are presented below, and a visual representation of the word content analysis across all interview transcripts is shown in the word cloud (Figure 1). The resultant themes and an outlining definition of each are summarised in Table 3. Within the reporting of the results, the words “client”, “customer”, and “horse owner” are used interchangeably.

### 3.1. Theme 1: Effective Communication


**“that lack of communication, people don’t realise the benefits of working together”**


Manner, quality, and consistency of communication with clients, other saddler fitters, and other professionals, respectively, were recognised to be pivotal to effective day-to-day interdisciplinary working. Clear lines of communication were acknowledged by all participants to be crucial to the accuracy of messaging between professionals, with direct contact between practitioners being essential to avoid misinformation, as highlighted by the comment below.


*I contacted said physio ……… she’s in the area, we know each other, we’ve talked in the past about previous horses. And it transpired that there was X, Y and Z going on with the horse and actually it was a Chinese whispers thing, what I’ve been told from the customer wasn’t actually what the physio had said to the customer. So, in actual fact having a relationship with that physio meant I could pick up the phone and say, what’s going on, and get it from the horse’s mouth…*


Despite complete agreement amongst participants that telephone calls were the ideal method of communication to ensure clarity between professionals, participants reported that they infrequently received direct calls from other professionals. This was presented as a barrier to effective communication; however, it appears the horse owner could be assumed to have relayed requests for other practitioners to call.


*I’d ask the customer to get them to phone me, …… quite often you might say that and no one will bother phoning. I don’t [know]… Um, one in ten [will telephone]*


Communication was considered by many to be essential to team working.


*I think it’s about knowing who else is involved, so who’s on your team, it’s about that team having good lines of communication and it’s about understanding what you’re supposed to be looking at and what other people are supposed to be looking at.*


When asked about what creates a good outcome when working with horse owners and their horses, two-way communication (i.e., between horse owner and practitioner) across the MDT, including the SMSQSF, was highlighted as a significant component.


*If everyone communicates their point…. I think you should always take from other people rather than trying to push your opinions on them.*


### 3.2. Theme 2: Multidisciplinary Expectations


**“Clients that want the best for their horse, build that support network around them so they know that they’re getting the right set up for the horse”**


Client and professionals’ expectations of a multidisciplinary approach, the benefits and challenges, and how a team approach can be adopted formed the cornerstone of this theme, and there were differences in interviewee experience. However, meeting the expectations of clients, other professionals, and their own personal standards of service were important to all study participants. But on some occasions, horse owner expectations were difficult to fulfil as they were not closely aligned with the practicalities of the saddle fitting process.


*But I think the expectation from customers a lot of the time is that you get a saddle and that should be it and it should be perfect and you should never ever have a problem…. I think expectation versus reality is very different.*


On consideration of the expectations of inter-disciplinary team working, results were differentiated as some clients and other professionals had expectations and experience of a team approach whereas others did not. When asked about client expectations of a multi-disciplinary approach, one interviewee identified a discrepancy between client groups and the concurrent need to educate clients.


*I think for me with my customers, there’s a split, there’s a real divide there with people that are, that want it, know about it and are trying to achieve it, and people that have no idea. I should, we should be educating them…… You know, and that’s only because they don’t know, they don’t know that a physio alongside the vet, alongside that saddle fitter, alongside the farrier can actually make everything all so much nicer and easier, they genuinely don’t know that.*


A number of participants recognised that client expectation of a team-based approach from professionals is a growing phenomenon, and that the SMSQSFs could be influential in the development of this way of working.


*I think they’re beginning to expect this sort of thing, but it’s taking a while…. I’m trying to educate my clients.*


### 3.3. Theme 3: Horse Welfare


**“It’s for the welfare of the horse”**


The fundamental role of the SMSQSF in safeguarding horse welfare was apparent throughout the interviews. Participants clearly recognised how they could provide opportunities to secure horse welfare and promote health longevity in ways which are unique to their profession.


*I think the biggest advantage is that what the vet and the physio will generally not see is the horse being ridden, and I think the saddle fitter and the instructor will observe the horse under saddle.*


Clear risks to horse welfare were highlighted if saddle fit is not given due consideration.


*[It is] partly because ……. horses are really, really forgiving and, you know, I think a common perception is my horse is not misbehaving, therefore I don’t need to get my saddle checked.*


Furthermore, respondents highlighted the need to work as a part of a cohesive team for paramount horse welfare.


*It’s for the welfare of the horse. It’s, the horse is, you know, everything is linked, it’s not just, you can’t just do a saddle fitting and think that that’s going to fix everything else that’s wrong with the horse. You know, you need everybody else there to have their input as well. And it’s all about making the horse as comfortable as possible, isn’t it?*


Effective multidisciplinary team working was considered to confer horse health benefits.


*The main advantage is a better looked after horse……. a happier, healthier horse with a rider that’s more balanced.*


### 3.4. Theme 4: Professional


**“You need to be professional”**


The theme of professionalism was substantial and accordingly encompassed three core sub-themes including: (i) professional boundaries; (ii) decision making; (iii) professional representation.

(i).Professional boundaries

Failure of other professionals to respect professional boundaries was a consistently reported challenge and indeed a potential source of conflict.


*It’s always trying to remind people that if you’re a qualified saddle fitter, you’re not a qualified physio so you can’t comment on any physio/veterinary issues but at the same time, the physio is not necessarily a qualified saddle fitter, so they shouldn’t really be commenting on saddle fit in any particular detail. If they don’t think it fits, then they should be saying, I don’t think it fits, or, I suggest you get your saddle fitter to check it. But they shouldn’t be saying, oh, this doesn’t fit because…*


This need for professional boundaries resonated with many participants.


*I think the first thing a multidisciplinary team has to do is, each person has to understand what their area of expertise is and where the boundaries are, and where they need to, whether they have information or they have an observation that would be beneficial to be shared with another member of that, that team.*


(ii).Decision making

Debate regarding who was responsible for instigating patterns of interdisciplinary working and promoting interdisciplinary working was an influential component of this theme, particularly in the context of veterinary referral. This was acknowledged as part of the SMSQSF decision-making process, but concerns were raised with respect to whom should undertake this responsibility.


*The horse in question, in my opinion, needed to be investigated by the vet…. as a saddle fitter it’s really hard to advise [on] veterinary intervention.*


(iii).Professional representation

The theme of professional representation incorporated recognition by other stakeholders and associated feelings of being undervalued as compared to other comparable professionals. Often, participants felt undervalued by both their clients and other professionals alike.


*…I don’t think that saddle fitting is seen as a profession in the same way that being a qualified physiotherapist or a qualified vet.*


Discussions around the professional representation across the sector were had, with participants concluding that they should be advocates for their own practice and better support each other.


*I’d probably seek advice from other fitters, or just to be able to, you know, to talk it through, see what their opinion is, if there’s anything else they’d try.*


### 3.5. Theme 5: Relationships


**“We, as professionals, have to put effort into cultivating relationships”**


The professional–client relationship was viewed to be fundamental to high-quality service. Client trust in their professional was seen to be linked to loyalty and repeat business. Equally, good relationships between all professionals was conducive to effective team working.


*Yes, I would say the majority of them [clients] are loyal, … So, it’s me they’re seeing and I like to think that I build quite a good relationship with most of my customers. You always get the odd person that is only out for the person that can, you know, see them the quickest or the cheapest, but I do have quite a regular customer base of people that come back to me, so I would like to think that they are loyal.*


Client relationship and rapport was viewed as pivotal to a successful team working outcome, particularly when veterinary involvement is required.


*…I have a particularly good relationship with this customer…. And because I had a good relationship with her, she was open to me talking [to the vet]*


Inter-professional relationships were viewed to be of equal importance and, indeed, crucial to effective team working.


*So, as I just mentioned before, you know, being able to put things into words for another professional to understand and to have that relationship and rapport.*


### 3.6. Theme 6: Working Together


**“One learns so much from other professionals to then take forwards”**


Across the participants, inter-disciplinary team working ranged from a daily occurrence to very infrequent experience, and the constituents of the team varied. Overall, participants indicated that the following professionals could contribute to the team: veterinarians, MSK practitioners, farriers, equestrian coaches (also referred to as “instructors” by participants), equine dental technicians, and other saddle fitters, and in some cases, the client (horse owner) considered themselves to be a part of the team. On occasions when team work was unsuccessful, some respondents highlighted how underrated the saddler fitter profession currently is.


*…. I think the first thing and the major thing is that I think saddle fitting in general is massively undervalued, that some people get it and some people don’t, and I think both owners and other practitioners don’t, don’t value saddle fitting, as being particularly, not, not, it’s not, important isn’t the word, but I don’t think they see it as relevant, as being as relevant as, you know, things that involve actually treating the horse like the vet and the physio would do.*


Furthermore, the practicalities of coordinating a team approach was a common thread.


*I have found that quite difficult to arrange a lot of the times, I suppose because around here we do cover such vast areas and we both, you know, I say both, me and whichever one of the others, are booked quite far in advance. It’s quite difficult to coordinate our diaries to do that, so normally it’s done by email or phone to be honest.*


There was recognition of a need for a clear protocol for team working.


*I think I’d like to see something a bit more official, because I never really know whether I’m supposed to discuss other people and other people’s horses with other practitioners, I don’t actually know what the proper protocol is.*


The financial constraints and additional costs to the client (horse owner) associated with team working were at the forefront of participants’ minds.


*I mean I think your biggest problem is the money, is the money.*



*…if the client can see pound notes flying out the window, they’re going to be a little bit reluctant*


Some participants had experienced reluctance on the part of the client to work with other professions.


*I do find resistance to any investigation by vets. They tend to be more amenable to [the] thoughts of physios.*


When participants were able to work directly alongside other practitioners, this was considered to be the gold standard.


*The advantages are obviously that you can get a much wider knowledge base providing possible modifications and suggesting possible outcomes for each modification. So, the advantage is more knowledge, obviously.*


However, they also recognised that there was work to be done to facilitate effective inter-disciplinary team working within this sector.


*Unless we have more understanding of one another’s areas, …we can’t work together. And I’m not saying that we all need to be an expert in every single field because that’s clearly not the case, and that’s clearly not possible.*


Education on the complexity of fellow professionals’ roles was believed to be a necessary precursor to more effective team working.


*Do we not think that word, education, is the key? Do we not, personally, again, I can only go from opinion, can’t I? But personally, unless, unless we have more understanding of one another’s areas, we can’t, we can’t work together.*


## 4. Discussion

This study used a qualitative approach to explore patterns of team working across the equestrian MDT. Throughout the interviews, discussion was forthcoming and engaged, indicative of strong sentiments on the opportunities and challenges associated with team working. Data obtained were rich, and six comprehensive themes were consistently evident. Cross cutting all themes, matters pertaining to communication were apparent as an important pillar in ensuring effective multidisciplinary teams. Communication with clients, i.e., horse owners, other professional stakeholders, and SMSQSF colleagues alike were all considered to be of equal importance. Direct communication between practitioners, without relying on the horse owner as an intermediary, was viewed as the “gold standard” by participants. Equally, communication barriers faced and opportunities taken or missed were consistently raised by the study participants. Whilst this topic has not been directly studied within equestrian MDT settings, the importance of quality communication is largely unsurprising given that it is already recognised as an essential component of high-performing interdisciplinary teams within animal healthcare [28,29].

All participants expressed an appreciation of the benefits of MDT for horses and horse owners alike. Findings from this study clearly indicate the fundamental and distinctive role of the SMSQSF in the ongoing improvements in the health and welfare status of the ridden horse, since they routinely and consistently see the horse and rider in the ridden environment. This requirement is embedded in the SMS Code of Conduct and provides an opportunity to signpost the horse owner to others within the MDT as required, for example, if there are concerns that saddle related issues may be linked to lameness or pain [17,18]. Participants gave examples where optimal saddle fit not only ensured the comfort of the horse during ridden activity but could enhance the performance of the horse–rider partnership, although the latter was perhaps less well recognised by others within the team. Within elite equestrian sport, the integration of interdisciplinary professionals is central to the concept of marginal gains theory, where performance is optimised by small improvements across a range of different areas, together producing more significant overall benefits [30]. Within human healthcare, there is recognition of the value of MDTs and collaborative working as a tool to continuously improve health standards [13,31,32]. This supports the adoption of an MDT approach within equine healthcare as a means of optimising the functionality and longevity of the horse–rider partnership outside of purely an elite performance context.

Further to the apparent benefits of an effective MDT for the horse–rider partnership, effective interdisciplinary teams can be advantageous to the practitioners themselves. The key benefits of learning from each other, providing support and guidance, and business-associated benefits were well reported within this study. This supports previous work [4] wherein a cohesive, mutual relationship between the vet and farrier was found to be beneficial to acquiring and maintaining clients, the ability to perform various actions that one profession is better equipped to provide than the other, and a constant source of experiential learning. These findings also reflect comparable research undertaken within human healthcare MDT settings, whereby the benefits of team working are indicated as improved care, enhanced knowledge, and good practice sharing [13,14]

Notwithstanding the opportunities conferred through effective interdisciplinary team working, pragmatic challenges were evident. Once again, whilst the practical challenges identified by participants are specific to the equestrian sector, there are parallels to effective team working in other professions, notably time and financial implications [33]. Limitations of the time resource of professionals, including diary constraints across members of the MDT, was highlighted as a significant impediment, with solutions having potential cost implications for horse owners.

Fundamental to the success of all MDTs is the structure and organisation of the team. In the equine sector, the horse owner is often the main decision maker. Accordingly, it could follow that the horse owner should be the MDT lead or be centremost to the team structure. Participants reported that some owners valued a team-based approach, as seen in previous studies [15,34,35], and hence were more likely to assume the role of lead. Other owners had little expectation of an MDT approach; hence, they were less motivated in this respect. Veterinary regulation within the animal health system in the UK inherently places the veterinarian as the team lead, whereas in human healthcare, there is recognition that no one healthcare professional holds the key to all patient needs [13]. There are, however, circumstances within animal healthcare wherein the team structure is more fluid and dynamic. In livestock systems, for example, the farmer assumes the role of MDT lead or acts as the centremost hub [36], and the team approach is strongly advocated [37,38]. The standard relationship between veterinarian and client may not always apply in an elite equestrian sport context, when the veterinarian needs to relate not only to the horse owner but also to a trainer, a rider, or even a team manager or selectors [39]. Human health professionals recognise and value the benefits of team working but putting the ideal into practice is not without practical and inter-professional challenges and barriers. Hierarchical issues [40], leadership culture, and professional status [13] are key challenges to effective teamwork. Extrapolating from the extensive literature on team working in human healthcare suggests that adoption of a team-based approach within equine healthcare and equestrian performance would be likely to face equivalent challenges, as reflected by the results of this study. However, recognition of the fundamental role and influence of the horse owner in the MDT approach and, indeed, decision making in general appears to be unavoidable and essential to the achievement of best practice.

Within this present study, positive relationships between all contributing stakeholders were indicated as an essential component of fruitful interdisciplinary working. The success of the relationships between equestrian professionals within the MDT was influenced by factors such as human behaviour, social and professional status, qualifications, and affiliations. The perceived deficit in the professional recognition and status of the SMSQSF practitioner was reported to influence participant’s confidence in building relationships with other professional stakeholders. The recently introduced “Equine Fitters Council”, with an associated directory of professional “fitters” within the United Kingdom, aims to “…promote education and encourage professional practice within the industry” [9] and should therefore have a positive impact on the professional status of “fitting”.

Participants reported a lack of awareness of the SMSQSF within the wider team, purported to lead to the encroachment of professional boundaries (by other paraprofessionals), with negative consequences for professional relationships. Communication and relationships were, unsurprisingly, inextricably linked, with one serving as a precursor to the other. SMSQSFs felt more enabled to use a direct approach if there was a pre-existing relationship. Some study participants reported that their attempts to communicate were often not reciprocated, which could deter them or others from making the effort in future. Findings also indicated that lines of communication could be improved or hampered through the input of key opinion leaders, such as other horse owners or yard managers or influencers with a proxy online presence.

This study highlights three areas relating to successful MDT working worthy of further research:(1)Greater understanding of how a team-based approach impacts horse welfare: Given that professions other than saddle fitters have the opportunity to influence saddle fit, better understanding of how all those within the MDT currently engage with horse owners could inform strategies for more effective MDT engagement on saddle fit matters. Indeed, the study of how equestrian professionals engage with each other and the horse owner on saddle fit matters, could be used as a model for improvements in MDT working on other equine health and welfare issues. As seen in human healthcare MDTs, an appreciation and recognition of the professional scope of practitioners by all professions within the team is essential to successful MDT working.(2)How to build effective inter-disciplinary relationships and communication across flexible teams: Each profession knowing when to consult and refer to others within the team aids inter-disciplinary relationships, so “professional relationships” should feature in the education programmes of those within the MDT. Recent work suggests that in a veterinary education environment, peer-assisted learning can be successful [41]. Practical strategies to provide opportunity for cross-disciplinary education programmes may be welcomed by those within MDTs and foster more-effective learning and greater cooperation between professions. This study indicated practical barriers centred on resources, including time and money, and effective modes of communication as challenges to a successful MDT-based approach. Some participants gave examples of using social media platforms such as WhatsApp groups to share information. The potential benefits and pitfalls of utilizing digital spaces to safely share images and video and to communicate with other professionals and the horse owner is worthy of investigation in an increasingly time-pressured working environment.(3)The role of the horse owner within the MDT: Acceptance of the horse owner as an essential contributor to the MDT is considered fundamental to a successful MDT but complex and differentiated according to the situation, the horse owner, and the professionals involved. This work suggests that there may be gaps in some horse owners’ understanding of the significance of saddle fit to horse welfare or misinformation provided by unregulated sources. The person taking on the leadership or central role in the MDT should have a high degree of knowledge. Education of horse owners by reliable sources may be a useful step towards increased owner awareness and responsibility for saddle fit. Again, effective, targeted, and evidence-based social media programmes have potential to be leveraged to this effect [42]. The role of the horse owner within the MDT; whether or not they should assume greater responsibility; and the barriers, challenges, and opportunities of the horse owner in a leadership role within the MDT comprise an exciting area for further work and one with potential for outcomes with practical implications for positive welfare outcomes for the ridden horse.

This exploratory study has opened the conversation regarding novel approaches to inter-professional working patterns across the UK equine sector. Further research is warranted on how an MDT approach may work in practice, where policy will support or challenge this model of care, and what opportunities team working creates in the continued quest for enhanced equine welfare.

## 5. Conclusions

As a contributing factor to horse welfare, saddlery fitting warrants professional consideration. Further study from the perspective of other professionals and horse owners would provide greater clarity on the barriers to, and opportunities for, developing effective multi-disciplinary teams. The effectiveness of the equestrian MDT in achieving and maintaining optimal saddlery fit for the horse is influenced by interdisciplinary communication, horse owner education, and the continued development of “fitting” as a profession.

## Figures and Tables

**Figure 1 animals-14-00559-f001:**
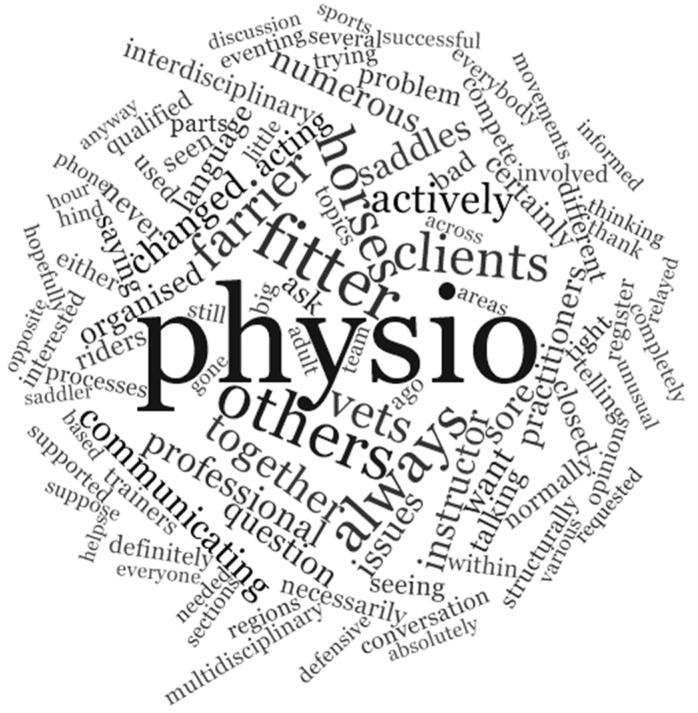
Visual representation of the word content analysis.

**Table 1 animals-14-00559-t001:** Glossary of terms.

Qualified Saddle Fitters (SMSQSF)	Society of Master Saddlers Qualified Saddle Fitters
Master Saddle Fitter (SMSMSF)	SMS master saddle fitters will hold the Society saddle fitting qualification, will have a minimum of seven years’ experience and will either be a member in their own right or will be employed by a member. They will also either be a Master Saddler or hold the Society’s Flocking & Flocking Adjustment qualification and will have provided references from both customers and fellow saddle fitters.
Master Saddler (SMSMS)	Master Saddlers are trained, skilled and qualified in their own right to make and repair saddlery. They will have a minimum of 7 years’ experience in the trade. Master Saddlers do not necessarily have retail premises.
Musculoskeletal practitioner	Musculoskeletal practitioners include animal/veterinary physiotherapists, animal chiropractors, animal osteopaths and animal massage therapists.
Multidisciplinary team	Professionals from different disciplines working in collaboration for a common goal

**Table 2 animals-14-00559-t002:** Summary of interview topics and areas explored.

Interview Topics	Areas Explored
Client Base	The nature of the participant’s business, client base, geographical area covered and client type
Understanding of ‘interdisciplinary’ and ‘multidisciplinary’	The nature of interdisciplinary and MDT practice for the participant, the frequency and nature of interactions with others, the process itself, channels of communication with others
Experiences of working with others	Client expectations of the MDT, advantages and disadvantages of working within a MDT, using specific examples
Future Development	The value of MDT working to the profession, barriers and possible improvements

**Table 3 animals-14-00559-t003:** Summary of themes.

Theme	Definition
1. Effective Communication	Quality and respectful communications are pivotal to successful inter-disciplinary interactions
2. Multidisciplinary Expectations	Potential for successful team-based approaches
3. Horse Welfare	Welfare as the centre-point of all professional interventions
4. Professionalism	Recognition and appreciation of professional recognition and boundaries, and the association with robust decision-making skills
5. Relationships	Value of positive and transparent client: professional and professional: professional relationships
6. Working Together	Benefits and challenges associated with a collaborative approach

## Data Availability

Anonymised data available on request.

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
