# Peer review of "Experiences of Interdisciplinary Working from the Perspective of the Society of Master Saddlers Qualified Saddle Fitters"

_animals, 2024, doi:10.3390/ani14040559_

Round 1

Reviewer 1 Report

Comments and Suggestions for Authors

I have reviewed the article "Experiences of interdisciplinary working from the perspective of Society of Master Saddlers’ Qualified Saddle Fitters." I enjoyed very much reading this paper, is well-written, the results are well-written and support your research. Generally, I recommend to remove the spaces between the same paragraphs because its adds confusion. Moreover, consider to check the formatting style.

In addition, you should address some specific comments for further clarity as below:

Line 56: I would suggest you to write United Kingdom

Why in the introduction there are spaces between the paragraphs?

Lines 55-59: I would suggest you in this part here to briefly mention the typical challenges faced by horse owners in assembling their multi-disciplinary team (MDT), to set the stage for the study's relevance.

Definition of MDT

Lines 66-69: The mention of the lack of evidence exploring MDTs in equestrian settings is a good lead-in to the paper's objective. Perhaps you should consider to include a sentence on why this gap is significant in terms of horse welfare and equestrian practices.

Lines 382-393: In these lines here it might be beneficial to include specific strategies or examples of how communication can be effectively managed, especially in challenging situations.

Lines 395-411:  You should consider elaborating on how this role can be more recognized and valued within the MDT, which could lead to better horse and rider outcomes.

Lines 414-429: In this part here, I highly recommend you in adding a comparison with challenges faced in human healthcare MDTs could provide a broader context an validate the study’s findings.

Lines 431-455: This section here could be enhanced by discussing how the leadership role can be more effectively distributed among team members, including the horse owner.

Lines 456-484:  You might consider providing recommendations on how to improve the professional status of SMSQSF within the equestrian community.

You should consider to add a new paragraph in which you should discuss the implications of your findings for future research or practical applications in the field.

In  your discussion section I highly recommend  to consider the insights from a recent study (DOI: 10.1016/j.jevs.2023.104537). The integration of these findings into your current discussion would not only complement the themes already explored but also provide a broader perspective on the practical application of these skills in veterinary education, particularly in the context of equine nutrition and healthcare. This study offers a valuable addition to the discourse on innovative veterinary education methods that can be used among veterinary professionals as well.

In light of the ubiquity of fake news, particularly in the realm of animal science, emphasizing the need to share your findings through contemporary channels such as social media becomes paramount. Consider elaborating on the potential impact of your research on public understanding and awareness. Discuss how leveraging platforms like Twitter, Facebook, Instagram or other social media outlets can contribute to the accurate dissemination of information, reaching a wider audience beyond the academic and scientific community.

Please see: 10.3390/ani13223503

Highlighting the importance of transparent communication and engaging the public directly through accessible channels will not only enhance the credibility of your research but also foster a better-informed society. By embracing modern communication tools, you can actively combat the spread of misinformation and ensure that your valuable contributions make a meaningful impact in both academic and public spheres.

Reviewer 2 Report

Comments and Suggestions for Authors

Dear Authors,

I thoroughly enjoyed reading your paper. It is eloquently written, presents its findings effectively, and makes a significant contribution to the field. The depth of analysis and the clarity of your arguments are particularly commendable. I would like to remind you, however, of the importance of adhering to the journal's formatting guidelines. Throughout the paper, I noticed several instances of irregular spacing. This could detract from the professional appearance of the manuscript and may not align with the journal's formatting standards. I suggest a careful review of the entire document to remove any unnecessary spaces and ensure that the formatting is consistent with the journal's requirements. These are relatively minor revisions but are essential for maintaining the high standards of publication. Once these adjustments are made, I believe the paper will be an excellent fit for the journal.

Specific comments:

Lines 61-65: The comparison with human healthcare MDTs is insightful. However, in my opinion you should consider to add a brief explanation of how this analogy helps in understanding equine MDTs for clarity.

Line 119: ensure that "SMSQSF’s" is correctly written as "SMSQSFs" (plural without the apostrophe).

Lines 382-392: I would suggest you to emphasize how your findings contribute to or differ from existing literature on team working in equestrian settings.

Lines 394-411: In this part here, you should consider expanding on how these roles can be better integrated or recognized within the broader MDT.

Lines 423-429: Addressing the practical challenges of interdisciplinary work is crucial. However, you should consider to suggest potential solutions or strategies to mitigate these challenges, based on your findings or existing literature.

Suggested Comment for Integration: I noticed in your discussion on the role of SMSQSF within the MDT (refer to the specific section or paragraph if possible), there's an emphasis on the importance of professional development and effective communication skills. In this context, it might be beneficial to reference a study that explores innovative educational strategies in a related field. Specifically, [https://doi.org/10.1016/j.jevs.2023.104537] investigates the flipped classroom and peer-assisted learning approach in veterinary education, which has shown promise in enhancing student engagement and learning outcomes. Integrating this reference could provide valuable insights into how such educational methodologies could be adapted for the training and continued professional development of equestrian MDT members, potentially improving interdisciplinary communication and collaboration.

General comment for integration: You should consider to add a new paragraph in which you should address any limitation of your study and suggest how they might be overcome in future research.  In addition, you should consider to address the implications of your findings for practice, policy, and future research. Consider the broader implications of your findings for the field of equestrian care and MDT working.

Round 2

Reviewer 1 Report

Comments and Suggestions for Authors

Dear Authors,

I wanted to extend my heartfelt congratulations to you and your team for the outstanding job you've done in revising your paper. I am genuinely impressed by the way you have meticulously incorporated the suggested revisions. Your commitment to improving the article's quality is evident, and I must say that the final result is nothing short of exceptional. The transformation from the initial draft to the current version is remarkable and a testament to your dedication to excellence.

Reviewer 2 Report

Comments and Suggestions for Authors

The authors have diligently addressed the review comments, significantly enhancing the paper's quality. As a result, it is now well-suited for publication.